

# Effects of food-based enrichment on enclosure use and behavioral patterns in captive mammalian predators: a case study from an Austrian wildlife park

Verena Puehringer-Sturmayr[1,2], Monika Fiby[3], Stephanie Bachmann[2], Stefanie Filz[2], Isabella Grassmann[2], Theresa Hoi[2], Claudia Janiczek[2] and Didone Frigerio[1,2]

[1] Department of Behavioral and Cognitive Biology, University of Vienna, Vienna, Austria
[2] Konrad Lorenz Research Center for Behavior and Cognition, University of Vienna, Gruenau im Almtal, Austria
[3] Zoo Design and Consulting, Vienna, Austria

## ABSTRACT

**Background**. Combining naturalistic enclosure design and animal welfare with visitor interests and education can be challenging for zoos and wildlife parks. To accomplish both purposes, different types of enrichment (food-based or non-food-based items, such as environmental, sensory, cognitive, social) can be used. The aim of the present study is to investigate the effect of food-based and olfactory enrichments on enclosure use, behavior, and visibility of captive brown bears (*Ursus arctos*), pine martens (*Martes martes*), domestic ferrets (*Mustela putorius furo*), and golden jackals (*Canis aureus*).
**Methods**. We used observational approaches to measure enclosure use, behavior, and visibility during three different experimental phases: (1) pre-enrichment (baseline, no experience with the enrichment yet), (2) during enrichment (enrichment was provided at low frequented locations in the enclosures that are easily visible to visitors), and (3) post-enrichment (enrichment was removed from the enclosures).
**Results**. We found that enrichment led to a uniform use of the enclosure and enhanced visibility in brown bears, increased activity budgets in pine martens, and observed high object interaction in both species. No effects of enrichment were detected in domestic ferrets. Golden jackals did not leave their burrows during daytime during the entire observation period; thus, observations were not possible at all. Our results suggest different effects of food-based enrichment, *e.g.*, enclosure use, temporal activity patterns, and animal visibility. However, further studies should control for the specific role of the factors involved. Our study represents one of the first explorations of food-based enrichment in rather understudied species.

# INTRODUCTION

In captivity, enclosures often lack biological and ecological conditions that can be found in the natural habitat of the animals. For instance, zoo-housed animals frequently have less

Corresponding author
Didone Frigerio,
didone.frigerio@univie.ac.at

space available than they would use in the wild and lack enclosure complexity, which are both correlated with negative welfare indices, such as stereotypies (*Carlstead, 1998*; *Clubb & Mason, 2003*; *Rose, Nash & Riley, 2017*; *Mellor, Brilot & Collins, 2018*). Among other approaches, environmental enrichment is most commonly used to reduce and prevent the development of abnormal behaviors, to stimulate activity, and to enhance quality of life for the animals (*Newberry, 1995*; *Mellen & Sevenich MacPhee, 2001*; *Mason et al., 2007*; *Fernandez, 2022*). Enrichments are classified into non-food-based items, such as environmental (*e.g.*, water element, climbing structure), sensory (*e.g.*, mirrors, scratch poles), cognitive (*e.g.*, novel object, puzzle feeder), or social (*e.g.*, conspecifics, mixed species enclosures, human–animal interactions) stimuli, and food-based items (*Swaisgood & Shepherdson, 2005*). However, enrichment is not only the stimulus/event, but also the interaction between the stimulus/event and a positive welfare change, such as improved behavior (*Fernandez, 2022*). While examinations of environmental enrichment are not necessarily novel, there have been few empirical investigations into factors and effects prior to, during and after the introduction of enrichment, and many captive species are still understudied.

Enrichment can stimulate animals' activities in previously underused enclosure areas, thereby allowing an efficient use of the available space and promoting the uniform use of different zones (*Renner & Lussier, 2002*; *Lawrence, Sherwen & Larsen, 2021*). As different zones should serve different purposes—*e.g.*, sleeping, feeding, bathing, reproduction, *etc.*— all zones should be used by the animals, although not neccessarily equally. Consequently, unused zones would be considered inadequate or of limited value (*De Azevedo et al., 2023*). For instance, brown bears (*Ursus arctos*; *Soriano, Vinyoles & Maté, 2015*) and Indian Leopards (*Panthera pardus; Mallapur, Qureshi & Chellam, 2010*) showed a higher use of environmentally enriched zones within their enclosure during enrichment, and thereby increased the use of certain areas, rather than using only a few of them. Similar findings were found in Asiatic lions (*Panthera leo persica*), which showed uniform enclosure use post-enrichment (*Goswami et al., 2021*). The extent to which the space of an enclosure is used is considered an indicator for the animal's welfare (*Ross et al., 2009*). Naturalistic environments reflect the respective home range size in the wild and offer hiding places for the animals, which is often accompanied by low animal visibility for the visitors (*Bitgood, Patterson & Benefield, 1988*). By contrast, other studies claim that enrichment increases the visibility of captive animals (*Foerder, Swanson & Collins, 2020*). Both aspects, *i.e.*, high animal visibility and the possibility of observing a species' behaviors (*e.g.*, social interactions, locomotion, grooming) foster and maintain visitor interest (*Bitgood, Patterson & Benefield, 1988*; *Kirchgessner & Sewall, 2015*) and thus can promote public education, enable research and conservation efforts (*Fernandez et al., 2009*; *Kuhar et al., 2010*).

In order to stimulate natural species-typical behavioral patterns, both the type and the timing when an enrichment is introduced are important (*Kuczaj et al., 2002*). In fact, zoo-housed animals often show very different behavioral patterns in comparison with individuals of the same species living in the wild (*e.g.*, Malayan sun bears (*Helarctos malayanus*), *Schneider, Nogge & Kolter, 2014*; wild chimpanzees (*Pan troglodytes*), *Inoue & Shimada, 2020*). For instance, explorative behavior related to acquisition and consumption

of food might be remarkably absent in zoos. Western lowland gorillas (*Gorilla gorilla gorilla*) in human care increased their activity budget reminiscent of individuals observed in the wild when feeding schedules varied, rather than providing food at fixed times (*Charmoy, Sullivan & Miller, 2015*). Similar findings were reported for cheetahs (*Acinonyx jubatus*), when feeding schedules, feeding locations, and olfactory enrichment varied (*Quirke & Riordan, 2011*). Determining temporal behavior patterns and activity budgets in captivity is particularly relevant for species that would allocate most of their active time to food procurement in the wild.

In addition, the diversity of the natural behavioral repertoire displayed by a species in captivity may increase when introducing enrichment to the enclosure (*Swaisgood & Shepherdson, 2005*; *McPhee & Carlstead, 2010*). Spectacled bears (*Tremarctos ornatus*) displayed more behaviors similar to their wild counterparts after climbing structures and other environmental items were introduced (*Renner & Lussier, 2002*) and stereotypic behavior decreased during the introduction of enrichment as compared to pre- and post-enrichment in captive sun bears (*Helarctos malayanus, Abdul-Mawah et al., 2022*). Furthermore, forms of food-based enrichment, including the use of novel objects to intensify the difficulty and time to acquire food, can increase the behavioral repertoire of captive animals. Kodiak (*Ursus arctos middendorffi*), polar (*Ursus maritimus*), and Asiatic black bears (*Ursus thibetanus*) were provided with ice treats (*i.e.,* ice blocks filled with food), which resulted in higher activity and reduced presence of abnormal behavior during enrichment sessions (*Forthman et al., 1992*). Furthermore, when European wolves (*Canis lupus lupus*) were exposed to food hidden in pouches or plastic food balls in the enclosure, stereotypic behaviors and negative social behaviors decreased, and they exhibited more exploratory behavior (*Riggio et al., 2019*). Lack of space or lack of enrichment may have detrimental effects on animal behavior, such as fewer social interactions (*e.g.,* chimpanzees (*Pan troglodytes*), *Koyama & Aureli, 2019*) or even the occurrence of stereotypic behaviors (*e.g.,* carnivores, ungulates, rodents, and primates, *Mason et al., 2007*; chickens (*Gallus gallus*), *Dixon, Duncan & Mason, 2010*). Natural species-specific behaviors are, however, necessary for fitness and consequently for zoos' conservation efforts for endangered species. Furthermore, zoo visitors may perceive abnormal behaviors as the animal being 'unhappy' or 'unhealthy' (*McPhee & Carlstead, 2010*), which could have a negative impact on the educational message of the zoo (*Fernandez et al., 2009*; *Godinez & Fernandez, 2019*). Thus, displaying natural species-typical behaviors is not only crucial for animal welfare, but also for the zoo management in order to maintain visitors' interest and satisfaction (*McPhee & Carlstead, 2010*).

The aim of this study was to determine the effect of enrichment (food-based and sensory items, *i.e.,* olfactory) on enclosure use, behavior and visibility (*i.e.,* number of individuals visible at a moment) of four carnivorous mammalian species in human care in a wildlife park setting: brown bears (*Ursus arctos*), pine martens (*Martes martes*), domestic ferrets (*Mustela putorius furo*), and golden jackals (*Canis aureus*). Several reasons supported the choice of these four species, which was taken in coordination with the Wildlife Park management: first, all four species showed little and non-uniform use of enclosure space; second, visibility of the individuals was low in two of the enclosures (*i.e.,* golden jackals and

ferrets; Cumberland Wildlife Park, personal communication); and third, some carnivores (in particular species with large home ranges and long daily travel distances in the wild, such as some bear species) are suggested to have more difficulties to cope with captive conditions than others, which would negatively impact animal welfare (*Clubb & Mason, 2007*). We predicted that enrichment elicits (1) higher use of non-used/less frequented areas within the enclosure, (2) changes in behavior, (3) changes in temporal patterns regarding enclosure space use and behavior, (4) higher activity budget, (5) increased visibility (*i.e.,* number of individuals visible at a moment) of animals, and (6) that the changes from prediction 1 to 5 elicited by enrichment are still present in the post-enrichment phase, albeit at a lower intensity.

## MATERIALS & METHODS

### Ethical statement

This study complies with all current Austrian laws and regulations concerning the work with captive wildlife. All data were collected non-invasively under Animal Experiment License Nr. GZ2021-0.873.421 by the Austrian Federal Ministry for Science and Research. The authors adhere to the 'Guidelines for the use of animals in research' as published in Animal Behaviour (1991, 41, 183–186). We confirm that the owner of the land, the Duke of Cumberland, gave permission to conduct behavioral studies on his site. A partnership agreement was signed between the research institution (KLF) and the Cumberland Wildlife Park stating the possibility of doing research with captive animals.

### Field site and study animals

This study was conducted in collaboration with the Cumberland Wildlife Park in Grünau im Almtal (Austria, 47°48′N, 13°56′E). The Wildlife Park is a popular tourist attraction in Upper Austria and houses more than 40 animal species, with around 500 animals on an area that expands over 60 ha. Here, we focused on four species: (1) brown bears ($N = 2$, 1 adult female and 1 adult male), (2) pine martens ($N = 4$, 1 adult female, 1 adult male, 2 juveniles), (3) domestic ferrets ($N = 3$, 2 adult females and 1 adult male), and (4) golden jackals ($N = 2$, 1 adult female and 1 adult male). The enclosures (approximate enclosure sizes: brown bears 5,100 m$^2$, pine martens 60 m$^2$, domestic ferrets 30 m$^2$, golden jackals 900 m$^2$) are designed to imitate natural habitats and are equipped with natural substrates, like soil, vegetation, rocks and wood. The brown bears are housed in a communal enclosure together with European wolves. Their enclosure features caves, forest sites, a pond, and access to a river. The enclosures of the brown bears and golden jackals both have open tops, allowing other animals, such as common ravens (*Corvus corax*), to enter. The other two enclosures are completely closed.

### Study design

Enclosure use and behavioral data were collected from 9th September 2021 to 20th November 2021. The observation period was divided into three phases: (1) pre-enrichment (between 10 and 12 days long; baseline, no experience with the enrichment yet), (2) during enrichment (between 14 and 17 days long; enrichment was provided at locations

in the enclosures that were/are low frequented but also easily visible to visitors), and (3) post-enrichment (between 7 and 12 days long; enrichment was removed from the enclosures). The length of the phases varied between species because of different enrichment implementations adapted for each species and in order to collect a similar number of observations between all phases (due to restrictions in animal keepers and observer availability). The post-enrichment phase was excluded from the brown bear dataset, due to the start of hibernation. The locations the enrichment was provided in varied throughout the experiment to enhance the effect and keep the animals interested in the enrichment (see Fig. 1 for details of where the enrichment was placed). Observations (enclosure use, behavioral patterns, activity budget, animal visibility and visitor flow) were performed between 0800 AM and 0600 PM including the opening hours (1000 AM to 0400 PM) of the Wildlife Park at three different times of day: (1) 0800 AM till 1100 AM (morning), (2) 1100 AM till 0200 PM (midday), and (3) 0200 PM till 0600 PM (afternoon). In total, we collected 95 observations for the brown bears (31 h and 40 min, pre-enrichment = 17 h and 20 min, during enrichment = 12 h and 20 min, post-enrichment = 2 h; mean ± SD observations per day = 3 ± 2), 79 observations for the pine martens (26 h and 20 min, pre-enrichment = 7 h and 20 min, during enrichment = 9 h and 40 min, post-enrichment = 7 h and 20 min; mean ±SD observations per day = 2 ± 1), 109 observations for the domestic ferrets (36 h and 20 min, pre-enrichment = 5 h and 40 min, during enrichment = 6 h and 20 min, post-enrichment = 6 h and 40 min; mean ± SD observations per day = 2 ± 1), and 122 observations for the golden jackals (40 h and 40 min, pre-enrichment = 17 h and 20 min, during enrichment = 10 h and 20 min, post-enrichment = 13 h; mean ± SD observations per day = 3 ± 2). Observation lengths between phases varied because observer availability differed between phases. Observations were performed on group-level. To measure animal visibility (the number of visible animals within the enclosure) and visitor flow (the number of people facing the enclosures), observers stood at locations that were easily accessible for visitors. As the brown bear and golden jackal enclosures were too big to get an overview of the whole area from one point alone, several observation points were selected (three at the brown bears, two at the golden jackals). During each observation, the animals were observed by one observer at one of the given observation points. The selection of which observation point was used was randomized. One observation point was sufficient at the ferrets and pine martens. Interobserver reliability was trained before starting data collection by defining and discussing the variables (*i.e.,* behaviors) to be collected. At a practical level, this happened in front of the enclosures to be observed: trainees recorded the same behaviors as the trainers indicated during the same observation slot.

## Enrichment

The enrichment items provided to the different species were food-based (brown bear, pine marten, ferret, and golden jackal) and olfactory enrichment (golden jackal). The time and location of enrichment in the enclosure were determined after the 'pre-enrichment' phase, to adapt enrichment items to the needs of each species (*i.e.,* according to their enclosure space use and temporal behavioral pattern observed in phase 1) and to enhance animal visibility for the visitors. The enrichment was introduced in the morning between 0800 AM
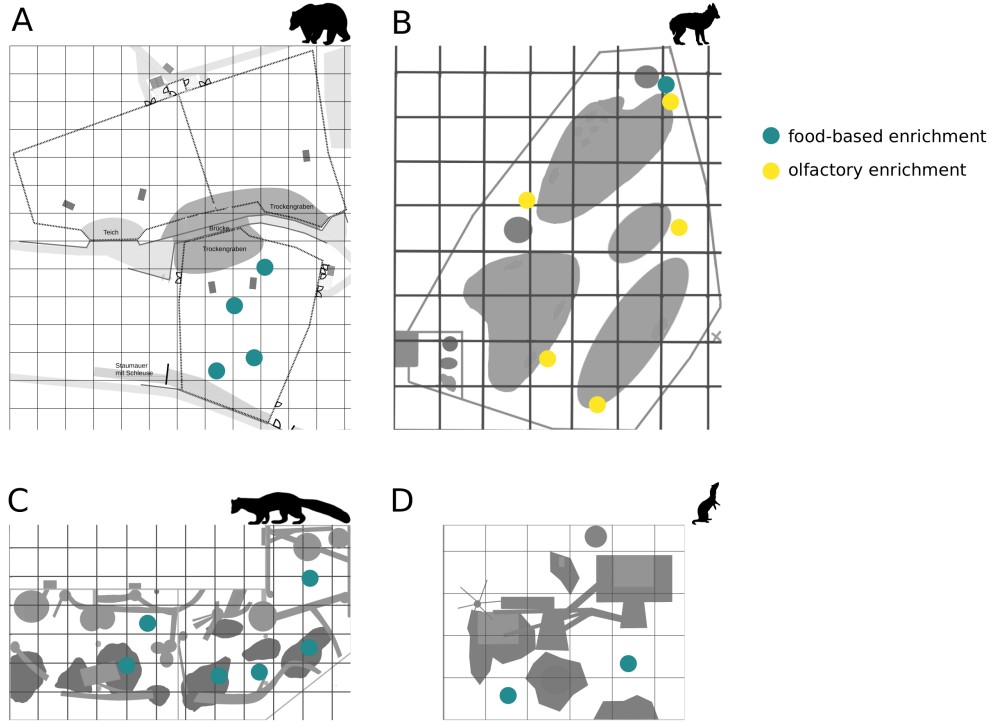

**Figure 1** **Location of enrichments.** The location of food-based enrichment in enclosures of (A) brown bears, (B) golden jackals, (C) pine martens, and (D) ferrets is marked in blue; yellow indicates location of the olfactory enrichment.

and 1100 AM, either additionally to the scheduled feeding (*i.e.,* brown bears) or outside the feeding context (*i.e.,* golden jackal, pine marten, and ferret). During feeding, the brown bears had to be confined in order to allow the animal caretakers to enter the enclosure and to distribute the food. This had to be done for safety reasons, which was not the case with the other three species. To avoid additional disturbances to the animals' temporal behavior patterns (additional confinements), we decided to coordinate the enrichment procedure with the feeding schedule in the brown bear enclosure. The enrichment was renewed every second day for the brown bears, pine martens, and ferrets, and once a week for the golden jackals. Enrichment was removed from the enclosures the evening before the day of renewal, to be renewed and placed in the enclosures the next morning. The timeline of the enrichment renewal was chosen to facilitate coordination with the animal keepers and for logistic reasons. The timing of enrichment (mainly regarding olfactory enrichment) presentation differed in golden jackals from that of the other species because scent does not dissipate too quickly as long as a strong scent is used and to avoid habituation of enrichment (*Clark & King, 2008*). Furthermore, as the golden jackals avoided the presence of humans from the beginning, we tried to limit disturbance by entering the enclosure and distributing food-based and olfactory enrichment to once a week. Observations were performed on all days.

There is evidence for brown bears in the wild spending about 50% of their active time with food procurement (*Seryodkin et al., 2013*; *Schneider, Nogge & Kolter, 2014*). Therefore, in our study, the brown bears received two 1 m long acacia logs as enrichment objects. Dried fruits, honey, yogurt and nuts were hidden in drilled holes of various sizes, which could be obtained by intensive and long-term activity. Food procurement consists largely of collecting and consuming various small food items, in order to ultimately meet their energy needs. By using the acacia logs, a natural behavior of food gathering is stimulated, that is often lost in zoo-housed animals due to fixed feeding procedures and schedules (*Grandia, Van Dijk & Koene, 2001*).

The pine martens and domestic ferrets were provided with an inter-crimped wire mesh (2 × 2 cm holes) covered open-topped wooden box measuring approximately 20 cm × 20 cm. The holes in the wire mesh were large enough for the animals to retrieve the food with their paws. The mesh was used to make the enrichment more challenging and to keep the animals interested in the enrichment. The box was filled with pine cones, bark, leaves, moss, cranberries, nuts, raisins, mealworms, and egg shells (egg shells were only provided to the ferrets) that the animals could reach with their paws. Pine martens and domestic ferrets can find this type of food in their natural environment, which they use to build up food stocks near their dens, especially in autumn (*i.e.,* the period of our data collection). The plant components can also be found by both species in forest habitats (*Marinis & Masseti, 1995*; *Bodey, Bearhop & McDonald, 2011*).

For the golden jackal, we used both food-based and olfactory enrichment. As food-based enrichment, we provided hollowed out pumpkins filled with insects, such as mealworms and stick insects. Golden jackals' diet in the wild is composed of small rodents, invertebrates, reptiles, birds, and plants (mainly fruits; *Giannatos et al., 2010*; *Markov & Lanszki, 2012*). Studies have shown that golden jackals feed on pumpkins (*Brooks, Haque & Ahmad, 1993*), making the combination of pumpkin and insects a good choice for enrichment (Monika Fiby, personal communication). Olfactory enrichment was included to stimulate enclosure exploration (*Clark & King, 2008*). We filled four jute bags (squares of about 50 cm × 50 cm) with either Przewalski horse (*Equus przewalskii*), European wolf, or rat feces (all species are housed in the Wildlife Park, the latter is used as food). The bags were randomly distributed in the enclosure and were attached to trees with twine (always two bags on the ground and two bags 1 m above the ground).

## Measuring enclosure use and behavioral patterns

All data were collected with the free iPad application Animal Observer (*Caillaud, 2016*, Animal Observer v1.0, https://fosseyfund.github.io/AOToolBox/). VP-S, SB, SF, IG, TH, and CJ performed the observations. Observations were collected on all days within the different experimental phases. We used instantaneous scan sampling (*Altman, 1974*; *Martin & Bateson, 2021*). The animals were observed over a time span of 20 min (randomized across species and the three time of day intervals), using four minutes intervals as sample points. Depending on species- and enclosure-size, the enclosures were divided into grids as follows: (1) 10 × 10 m grid for the brown bear enclosure, (2) 1 × 1 m grid for the domestic ferret and pine marten enclosures, and (3) 5 × 5 m grid for the golden jackal enclosure.
The squares of the grid were labeled with letters and numbers. Enclosure use was recorded by marking the x,y position of each individual at the end of each scan interval (*i.e.,* every four minutes) on the enclosure map using the app. Each x,y position was related to a square of the grid. Furthermore, behavioral data were collected as binomial variables for each individual within each scan sampling of the observation (*i.e.,* behavior was observed during the scan depicted as 1/not observed during the scan depicted as 0 for each individual in the enclosure): (1) behavior patterns, *i.e.,* self-maintenance and vigilance, locomotion (walking, pacing, climbing, hanging, approach/retreat), affiliative and agonistic interactions between conspecifics and heterospecifics, object (enrichment and other natural objects within the enclosure, such as stones, *etc.*) and human interaction and (2) activity budget, *i.e.,* activity in general (active, non-active). In addition, we recorded visibility (*i.e.,* number of animals visible at the moment of scan) and visitor flow at the moment of observation. Visitor flow was defined as the number of people facing the enclosures (*i.e.,* 0, <10, 10–20, >20 visitors at the enclosure) during observations.

## Data analysis

All analyses were performed using the software R 4.0.2 (*R Core Team, 2020*) and the package 'mgcv' (*Wood, 2011*). The 'post-enrichment' phase (data collected in November) was excluded from the brown bear dataset, as both brown bears were already showing signs of starting hibernation and, therefore, enclosure use and behavioral patterns were not comparable to the previous phases. The data of the golden jackals were entirely excluded from the statistical analysis, as both individuals did not leave their burrows during daytime during the entire observation period.

**Enclosure use.** The species' extent of enclosure use was assessed using the traditional Spread of Participation Index (SPI; *Dickens, 1955*), as we were interested in the change that enrichment elicits in enclosure use, rather than in the use of enclosure resources in general. The traditional SPI requires all enclosure zones to be of equal size. The SPI was calculated for each combination of phase and time of day (*i.e.,* pre-enrichment morning between 0800 AM and 1100 AM, pre-enrichment midday between 1100 AM and 0200 PM, pre-enrichment afternoon between 0200 PM and 0600 PM, enrichment morning between 0800 AM and 1100 AM, enrichment midday between 1100 AM–0200 PM, enrichment afternoon between 0200 PM and 0600 PM, post-enrichment morning between 0800 AM and 1100 AM, post-enrichment midday between 1100 AM–0200 PM, post-enrichment afternoon between 0200 PM and 0600 PM). SPI values range from 0 (equal use of the enclosure) to 1 (unequal use of the enclosure, *i.e.,* some enclosure zones are preferred over others). The formula for the traditional SPI is as follows: $SPI = M(n_b - n_a) + (F_a - F_b)/2(N - M)$; where N is the total number of observations made across all enclosure zones, M is the mean number of observations made per enclosure zone, $n_a$ is the number of zones that have a total number of observations greater than M, $n_b$ is the number of zones that have a total number of observations fewer than M, $F_a$ is the total number of observations in zones with frequencies greater than M, and $F_b$ is the total number of observations in zones with frequencies fewer than M.

**Temporal behavioral patterns and activity budget.** To investigate whether enrichment influences temporal behavior patterns (*i.e.*, self-maintenance and vigilance, locomotion, affiliative and agonistic interactions between conspecifics and heterospecifics, object and human interaction) and activity budget (active, non-active) we used separate generalized linear mixed models (GLMM) with binomial error structure and logit link function. We included phase, time of day, and visitor flow as fixed factors. Originally, we included also the interaction between phase and time of day in all models. However, some models did not converge, and therefore we excluded the interaction from the model. Observation ID was added as random effect to account for the possibility that behavioral patterns varied between scan samples. Each species and behavior were analyzed in separate models. Only observations where the individual behaviors could be observed were used for the final sample size. The samples for the brown bear models consisted of the following scan counts: 306 self-maintenance/other behavior scans (55 with self-maintenance), 306 locomotion/other behavior scans (46 with locomotion), 325 affiliative interaction/other behavior scans (two with affiliative interaction), 325 agonistic interaction/other behavior scans (one with agonistic interaction), 295 object interaction/other behavior scans (45 with object interaction), 325 human interaction/other behavior scans (one with human interaction), and 290 active/non-active scans (163 with activity).

The samples for the pine marten models consisted of the following scan counts: 959 self-maintenance/other behavior scans (45 with self-maintenance), 959 locomotion/other behavior scans (321 with locomotion), 948 affiliative interaction/other behavior scans (14 with affiliative interaction), 948 agonistic interaction/other behavior scans (68 with agonistic interaction), 948 object interaction/other behavior scans (71 with object interaction), 948 human interaction/other behavior scans (seven with human interaction), and 869 active/non-active scans (621 with activity).

The samples for the ferret models consisted of the following scan counts: 311 self-maintenance/other behavior scans (19 with self-maintenance), 311 locomotion/other behavior scans (60 with locomotion), 289 affiliative interaction/other behavior scans (two with affiliative interaction), 289 agonistic interaction/other behavior scans (zero with agonistic interaction), 289 object interaction/other behavior scans (five with object interaction), 289 human interaction/other behavior scans (five with human interaction), and 155 active/non-active scans (125 with activity).

**Animal Visibility.** To analyze how the proportion of visible animals varied with phase, time of day, and visitor flow (the number of people facing the enclosures) we used a GLMM with binomial error structure and logit link function. Observation ID and scan sampling ID nested in observation ID were included as random intercept. The reason for including this latter random intercept was that we had multiple scans per observation. In R such an analysis of proportions is possible by using a two-columns matrix with the number of visible and not visible individuals as the response.

Only observations where the individual behaviors could be observed were used for the final sample size. The samples for the brown bear models consisted of the following scan counts: 450 visibility scans (197 with visible animals).

The samples for the pine marten models consisted of the following scan counts: 468 visibility scans (398 with visible animals).

The samples for the ferret models consisted of the following scan counts: 642 visibility scans (124 with visible animals).

As an overall test of the effect of the fixed effects we compared the full model with a null model lacking the fixed effects but otherwise being identical to the full model using a likelihood ratio test.

To rule out collinearity we determined variance inflation factors (VIF) for a standard linear model excluding the random effect. No collinearity among phase, time of day and visitor flow was detected.

The models were fitted in R using the function glmer of the R package lme4. Tests of the individual fixed effects were derived using likelihood ratio tests; R function drop1 with argument 'test' set to 'Chisq'.

## RESULTS

Regarding the golden jackals, both individuals did not leave their burrows during daytime during the entire observation period. However, we know that the introduced enrichments were used during night, which was recorded with wildlife cameras. For this reason, no observations of golden jackals were possible and they were excluded from the statistical analysis.

**Enclosure use.** During enrichment, brown bears (Fig. 2) used more zones of the enclosure in the morning—in particular zones where enrichment was introduced—(SPI values: morning = 0.686, midday = 0.881, afternoon = 0.851), as compared to pre-enrichment (Figs. 2A–2B, SPI values: morning = 0.955, midday = 0.812, afternoon = 0.806; Fig. 3A). In contrast, enrichment did not elicit a change in enclosure use in pine martens (Fig. 4, SPI values: pre-enrichment –morning = 0.527, midday = 0.532, afternoon = 0.444; enrichment –morning = 0.518, midday = 0.530, afternoon = 0.496; post-enrichment –morning = 0.642, midday = 0.560, afternoon = 0.562; Fig. 3B) and domestic ferrets (Fig. 5, SPI values: pre-enrichment—morning = 0.567, midday = 0.672, afternoon = 0.612; enrichment—morning = 0.556, midday = 0.537, afternoon = 0.597; post-enrichment—morning = 0.567, midday = 0.505, afternoon = 0.672; Fig. 3C).

**Temporal behavioral patterns and activity budget.** Overall, there was a clear impact of the fixed effects phase, time of day, and visitor flow on object interaction (likelihood ratio test comparing full and null model: $X^2 = 24.025$, $df = 5$, $p < 0.001$) in brown bears. More specifically, object interaction in brown bears was more frequently observed during the morning as compared to midday and afternoon (marginal $R^2 = 0.49$; Table 1; Fig. 6). However, phase and visitor flow had no effect on object interaction. Phase, time of day, and visitor flow had no impact on activity (likelihood ratio test comparing full and null model: $X^2 = 10.579$, $df = 5$, $p < 0.060$), locomotion (likelihood ratio test comparing full and null model: $X^2 = 1.971$, $df = 5$, $p = 0.853$), and self-maintenance (likelihood ratio test comparing full and null model: $X^2 = 1.693$, $df = 5$, $p = 0.890$) in brown bears.

In pine martens, there was an overall effect of the fixed effects phase, time of day, and visitor flow on object interaction (likelihood ratio test comparing full and null model:

A

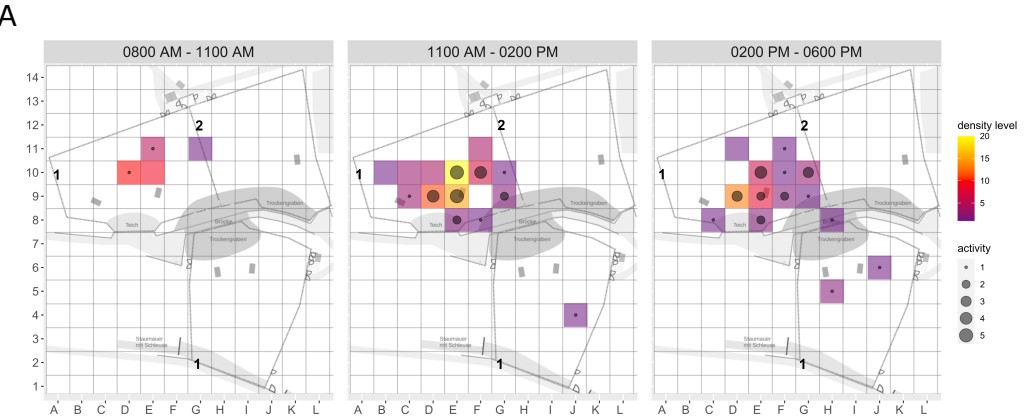

B

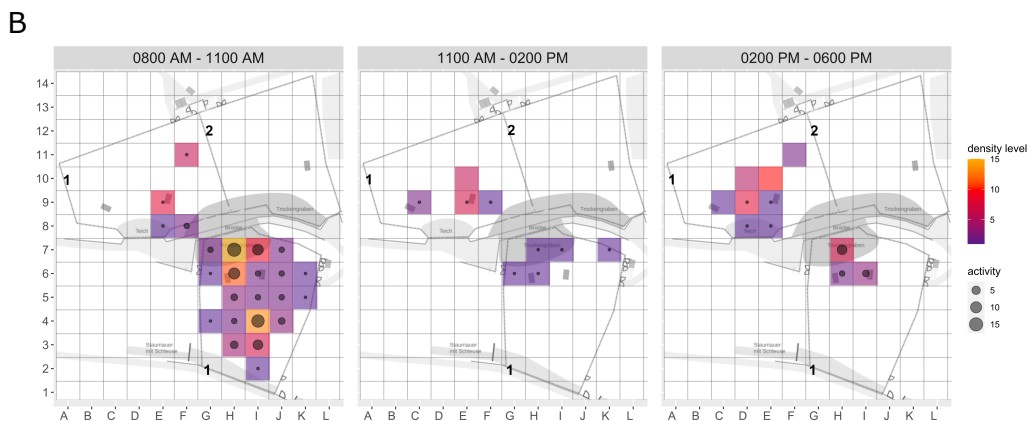

**Figure 2  Brown bear enclosure use.** The extent of enclosure use pre-enrichment (A) and during enrichment (B) is shown on a scale (density level) from purple (low) to yellow (high) for each square of the 10x10 m grid and divided into three time of day intervals (0800 AM –1100 AM, 1100 AM–0200 PM, and 0200 PM–0600 PM). Density level indicates the total number of sightings of individuals in a specific sector independent of activity level. Point size increases with activity level (*i.e.,* total number of sightings of active animals in a specific sector of the enclosure) per square. Enclosures labelled with '1' is accessible for brown bears and European wolves. Enclosure labelled with '2' is only accessible for the European wolves.

$X^2 = 19.524$, $df = 6$, $p = 0.003$), activity (likelihood ratio test comparing full and null model: $X^2 = 16.549$, $df = 6$, $p = 0.011$), locomotion (likelihood ratio test comparing full and null model: $X^2 = 35.560$, $df = 6$, $p < 0.001$), and agonistic interaction (likelihood ratio test comparing full and null model: $X^2 = 25.307$, $df = 6$, $p < 0.001$). Specifically, object interaction increased overall during enrichment and was generally higher in the morning and during midday as compared to the afternoon (Fig. 7A). Pine martens' activity (marginal $R^2 = 0.13$; Fig. 7B) and locomotion (marginal $R^2 = 0.14$; Fig. 7C) increased from pre-enrichment to enrichment to post-enrichment. Furthermore, agonistic interactions were observed less frequently during post-enrichment compared to the other phases (marginal $R^2 = 0.15$; Fig. 7D), and fewer agonistic interactions occurred when fewer than

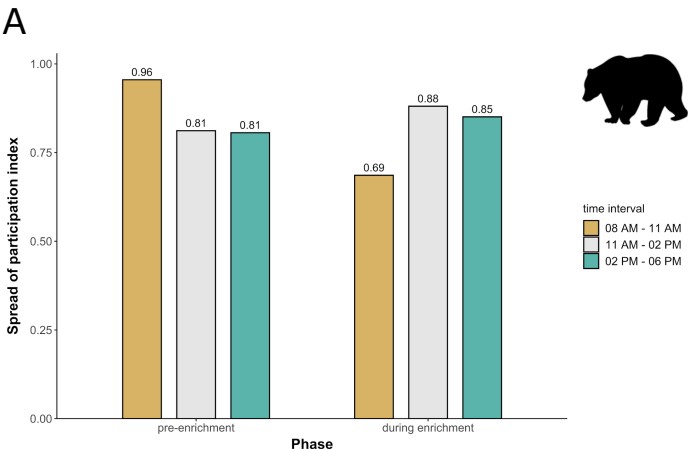

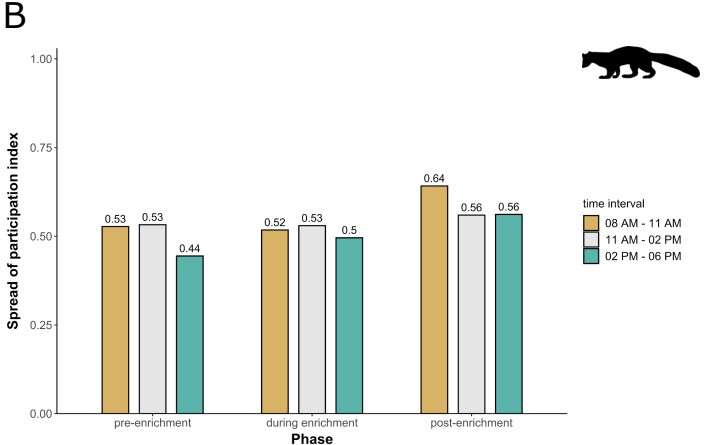

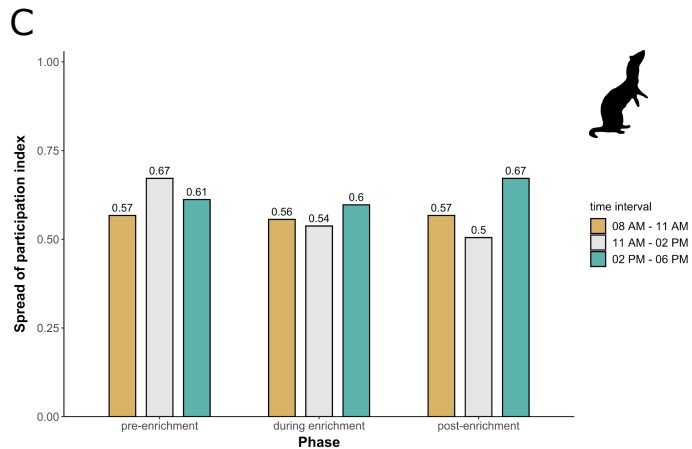

**Figure 3 Spread of participation index (SPI, enclosure use).** SPI is shown for (A) brown bears, (B) pine martens, and (C) ferrets, separately for each combination of phase (pre-enrichment, during enrichment, post-enrichment) and time of day interval (yellow: 0800 AM–1100 AM, grey: 1100 AM–0200 PM, blue: 0200 PM–0600 PM). SPI values range from 0 (equal use of the enclosure) to 1 (unequal use of the enclosure, *i.e.*, some enclosure zone are preferred over others).

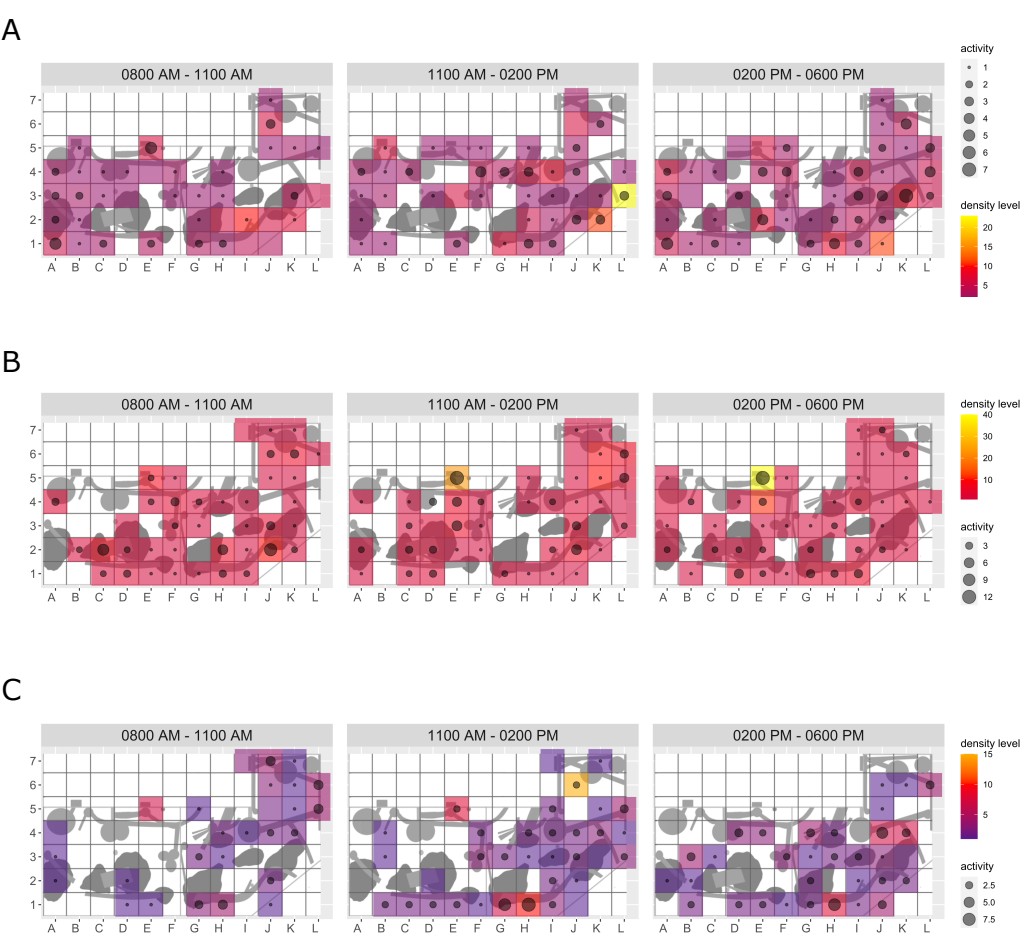

**Figure 4   Pine marten enclosure use.** The extent of enclosure use pre-enrichment (A), during enrichment (B), and post-enrichment (C) is shown on a scale (density level) from purple (low) to yellow (high) for each square of the 1 x 1 m grid and divided into three time of day intervals (0800 AM–1100 AM, 1100 AM–0200 PM, and 0200 PM–0600 PM). Density level indicates the total number of sightings of individuals in a specific sector independent of activity level. Point size increases with activity level (*i.e.,* total number of sightings of active animals in a specific sector of the enclosure) per square.

10 visitors were watching the enclosure as compared to 0 and 10-20 visitors (Table 1). There was no obvious difference between time of day and visitor flow with regard to activity and locomotion, and agonistic interactions did not differ between time of day intervals. The fixed factors phase, time of day, and visitor flow had no effect on self-maintenance (likelihood ratio test comparing full and null model: $X^2 = 2.689, df = 6, p = 0.847$) in pine martens.

Phase, time of day, and visitor flow had no impact on activity (likelihood ratio test comparing full and null model: $X^2 = 7.048, df = 5, p = 0.217$), locomotion (likelihood ratio test comparing full and null model: $X^2 = 4.534, df = 5, p = 0.475$), and self-maintenance (likelihood ratio test comparing full and null model: $X^2 = 9.809, df = 5, p = 0.081$) in domestic ferrets.

A

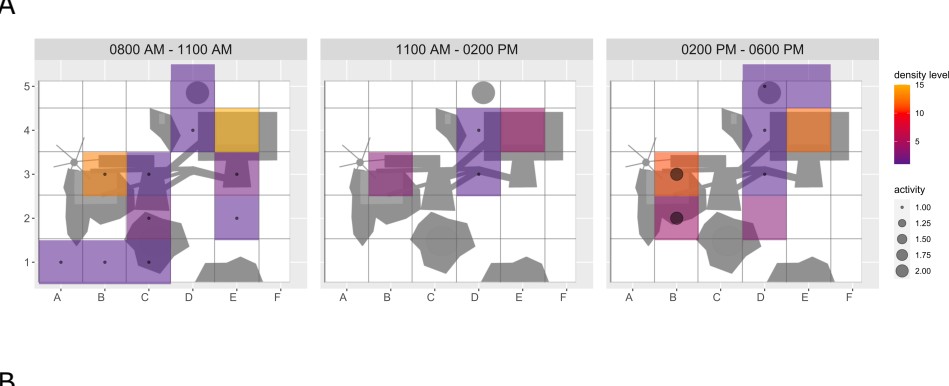

B

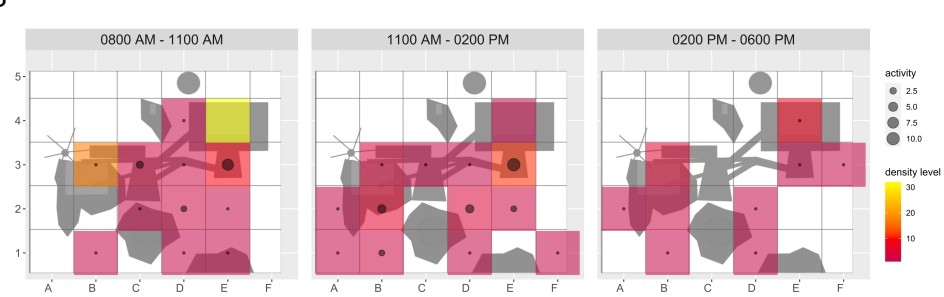

C

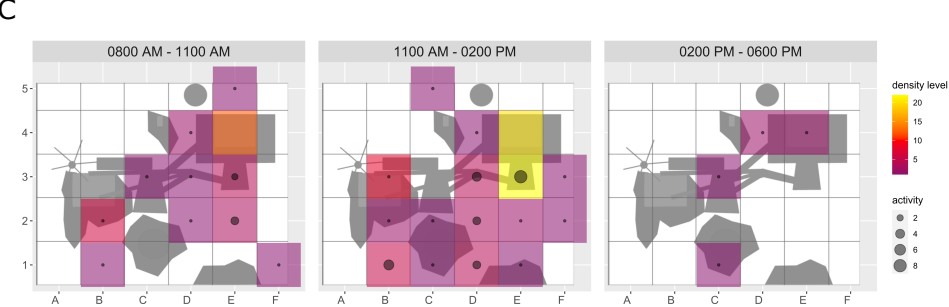

**Figure 5** **Domestic ferret enclosure use.** The extent of enclosure use pre-enrichment (A), during enrichment (B), and post-enrichment (C) is shown on a scale (density level) from purple (low) to yellow (high) for each square of the $1 \times 1$ m grid and divided into three time of day intervals. (0800 AM–1100 AM, 1100 AM–0200 PM, and 0200 PM–0600 PM) Density level indicates the total number of sightings of individuals in a specific sector independent of activity level. Point size increases with activity level (*i.e.,* total number of sightings of active animals in a specific sector of the enclosure) per square.

Some behaviors were observed rarely (less than 5% of the total observations; *e.g.,* affiliative and human interactions for all three species, agonistic interactions for brown bears and domestic ferrets, and object interactions for domestic ferrets) and were thus not included in the models.

**Visibility.** In brown bears, there was an overall effect of phase, time of day, and visitor flow on the number of individuals visible at the moment of scan (likelihood ratio test comparing full and null model: $X^2 = 17.177$, $df = 5$, $p = 0.004$; Fig. S1). Visibility

**Table 1  Results of the behavioral pattern models.** Estimates, together with standard errors (SE), confidence intervals (lower and upper CI), significance tests as well as minimum and maximum of model estimates obtained when excluding individual terms one at a time are given.

| Term | Estimate | SE | Lower CI | Upper CI | $\chi^2$ | df | p | min | max |
|---|---|---|---|---|---|---|---|---|---|
| **Brown bear object interaction** | | | | | | | | | |
| Intercept | −1.604 | 1.037 | −18.214 | 0.572 | | | [1] | −2.534 | −0.341 |
| Phase (during enrichment)[2] | 0.955 | 1.021 | −1.235 | 17.535 | 0.827 | 1 | 0.363 | −0.241 | 1.911 |
| Time interval (1100 AM–0200 PM)[3] | −2.424 | 1.065 | −64.351 | −0.900 | 15.069 | 2 | 0.001 | −3.128 | −1.838 |
| Time interval (0200 PM–0600 PM)[3] | −3.655 | 1.206 | −74.415 | −2.017 | | | | −22.912 | −3.389 |
| Visitor flow (<10)[4] | −0.808 | 0.878 | −13.350 | 0.904 | 1.569 | 2 | 0.456 | −1.245 | −0.314 |
| Visitor flow (10–20)[4] | 0.516 | 0.759 | −1.196 | 2.127 | | | | 0.085 | 0.866 |
| **Pine marten activity** | | | | | | | | | |
| Intercept | 0.677 | 0.494 | −0.275 | 1.686 | | | [1] | 0.488 | 0.949 |
| Phase (during enrichment)[2] | 1.632 | 0.516 | 0.713 | 2.660 | 12.455 | 2 | 0.002 | 1.460 | 1.787 |
| Phase (post-enrichment)[2] | 1.694 | 0.557 | 0.572 | 2.788 | | | | 1.518 | 1.835 |
| Time interval (1100 AM–0200 PM)[3] | −0.801 | 0.453 | −1.708 | 0.052 | 4.104 | 2 | 0.128 | −1.319 | −0.706 |
| Time interval (0200 PM–0600 PM)[3] | −0.010 | 0.537 | −1.027 | 1.089 | | | | −0.290 | 0.162 |
| Visitor flow (<10)[4] | −0.165 | 0.230 | −0.627 | 0.288 | 0.568 | 2 | 0.753 | −0.266 | −0.046 |
| Visitor flow (10–20)[4] | 0.190 | 1.014 | −1.665 | 12.760 | | | | −1.277 | 1.295 |
| **Pine marten locomotion** | | | | | | | | | |
| Intercept | −1.272 | 0.318 | −1.910 | −0.670 | | | [1] | −1.389 | −1.112 |
| Phase (during enrichment)[2] | 0.185 | 0.306 | −0.418 | 0.762 | 32.104 | 2 | <0.001 | 0.026 | 0.296 |
| Phase (post-enrichment)[2] | 1.805 | 0.336 | 1.204 | 2.492 | | | | 1.708 | 1.909 |
| Time interval (1100 AM–0200 PM)[3] | −0.070 | 0.296 | −0.632 | 0.562 | 2.837 | 2 | 0.242 | −0.193 | −0.003 |
| Time interval (0200 PM–0600 PM)[3] | 0.402 | 0.316 | −0.212 | 1.034 | | | | 0.296 | 0.475 |
| Visitor flow (<10)[4] | −0.116 | 0.197 | −0.529 | 0.263 | 0.835 | 2 | 0.659 | −0.184 | −0.033 |
| Visitor flow (10–20)[4] | −0.581 | 0.799 | −2.615 | 0.999 | | | | −14.256 | −0.017 |
| **Pine marten agonistic interaction** | | | | | | | | | |
| Intercept | −2.255 | 0.563 | −3.489 | −1.320 | | | [1] | −2.587 | −2.110 |
| Phase (during enrichment)[2] | 0.414 | 0.539 | −0.668 | 1.515 | 8.871 | 2 | 0.012 | 0.254 | 0.670 |
| Phase (post-enrichment)[2] | −1.401 | 0.674 | −3.154 | −0.093 | | | | −1.755 | −1.108 |
| Time interval (1100 AM–0200 PM)[3] | −0.760 | 0.525 | −1.857 | 0.303 | 2.131 | 2 | 0.345 | −0.933 | −0.567 |
| Time interval (0200 PM–0600 PM)[3] | −0.478 | 0.557 | −1.689 | 0.687 | | | | −0.733 | −0.262 |
| Visitor flow (<10)[4] | −1.485 | 0.422 | −2.597 | −0.739 | 17.061 | 2 | <0.001 | −2.291 | −1.220 |
| Visitor flow (10–20)[4] | 1.063 | 0.986 | −19.019 | 3.079 | | | | −13.784 | 2.968 |
| **Pine marten object interaction** | | | | | | | | | |
| Intercept | −4.923 | 0.989 | −18.442 | −3.325 | | | [1] | −5.451 | −4.632 |
| Phase (during enrichment)[2] | 3.539 | 0.903 | 1.958 | 16.923 | 17.164 | 2 | <0.001 | 3.306 | 4.128 |
| Phase (post-enrichment)[2] | 0.300 | 1.180 | −14.051 | 13.874 | | | | −0.479 | 0.997 |
| Time interval (1100 AM–0200 PM)[3] | −1.382 | 0.788 | −3.085 | 0.217 | 4.764 | 2 | 0.092 | −1.702 | −1.173 |
| Time interval (0200 PM–0600 PM)[3] | −1.918 | 0.878 | −4.097 | −0.206 | | | | −2.423 | −1.706 |
| Visitor flow (<10)[4] | 0.271 | 0.418 | −0.550 | 1.121 | 0.632 | 2 | 0.729 | 0.107 | 0.400 |
| Visitor flow (10–20)[4] | −12.903 | 323.817 | −16.060 | −8.563 | | | | −14.419 | −11.960 |

**Notes.**

[1] Not indicated because of having a very limited interpretation.

[2] Dummy coded with pre-enrichment being the reference category.

[3] Dummy coded with the time interval from 0800 AM till 1100 AM being the reference category.

[4] Dummy coded with 0 visitors being the reference category.

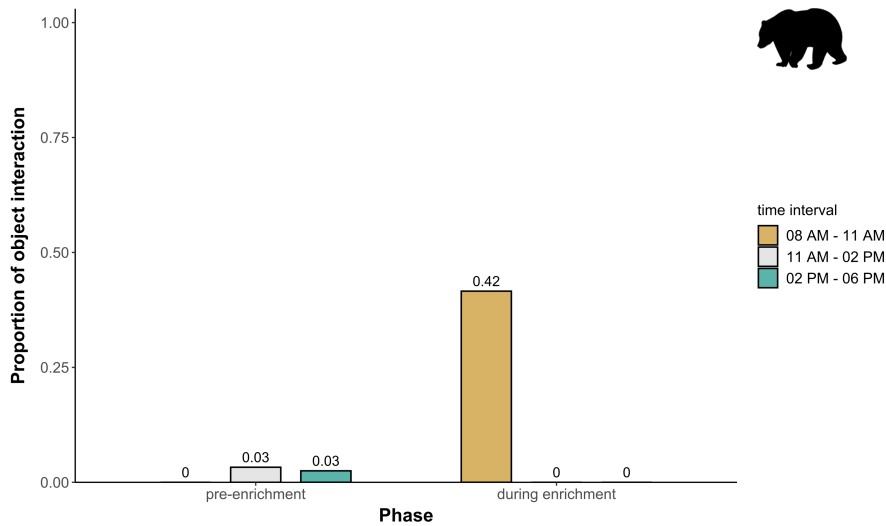

**Figure 6  Proportion of temporal behavioral patterns in brown bears.** The proportion of object interaction is shown separately for each combination of phase (pre-enrichment, during enrichment) and time of day interval (yellow: 0800 AM–1100 AM, grey: 1100 AM–0200 PM, blue: 0200 PM–0600 PM).

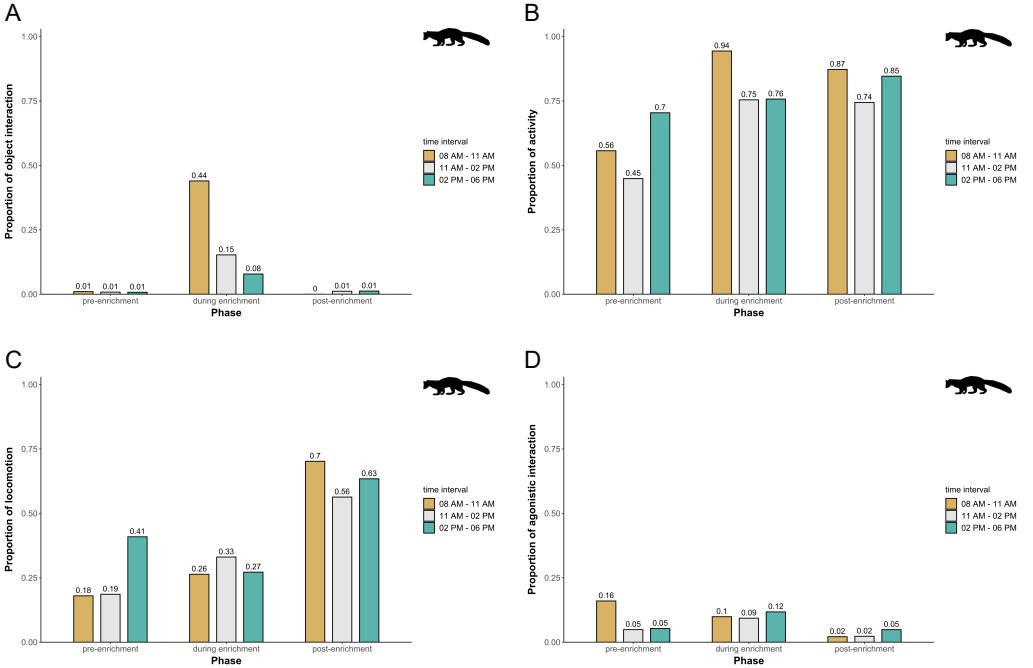

**Figure 7  Proportion of temporal behavioral patterns and activity budget in pine martens.** Shown are the proportions of object interactions (A), activity (B), locomotion (C), and agonistic interactions (D), separately for each combination of phase (pre-enrichment, during enrichment, post-enrichment) and time of day interval (yellow: 0800 AM–1100 AM, grey: 1100 AM–0200 PM, blue: 0200 PM–0600 PM).

**Table 2 Results of the visibility model.** Estimates, together with standard errors (SE), confidence intervals (lower and upper CI), significance tests as well as minimum and maximum of model estimates obtained when excluding individual terms one at a time are given.

| Term | Estimate | SE | Lower CI | Upper CI | $\chi^2$ | df | p | min | max |
|---|---|---|---|---|---|---|---|---|---|
| **Brown bear visibility** | | | | | | | | | |
| Intercept | −2.592 | 1.311 | −6.206 | −0.305 | | | [1] | −3.222 | −2.123 |
| Phase (during enrichment)[2] | 1.682 | 1.215 | −0.574 | 4.720 | 2.025 | 1 | 0.155 | 1.259 | 2.079 |
| Time interval (1100 AM–0200 PM)[3] | −1.750 | 1.395 | −5.115 | 0.626 | 1.671 | 2 | 0.434 | −2.215 | −1.283 |
| Time interval (0200 PM–0600 PM)[3] | −0.517 | 1.447 | −3.929 | 2.670 | | | | −0.946 | −0.048 |
| Visitor flow (<10)[4] | 1.223 | 0.404 | 0.433 | 2.112 | 13.479 | 2 | 0.001 | 0.924 | 1.433 |
| Visitor flow (10–20)[4] | 2.180 | 0.860 | 0.327 | 4.436 | | | | 1.744 | 3.291 |

Notes.
[1] Not indicated because of having a very limited interpretation.
[2] Dummy coded with pre-enrichment being the reference category.
[3] Dummy coded with the time interval from 0800 AM till 1100 AM being the reference category.
[4] Dummy coded with 0 visitors being the reference category.

increased with the number of visitors (marginal $R^2 = 0.09$; Table 2). Phase, time of day, and visitor flow had no obvious effect on the visibility of domestic ferrets (likelihood ratio test comparing full and null model: $X^2 = 8.323$, $df = 5$, $p = 0.139$) and pine martens (likelihood ratio test comparing full and null model: $X^2 = 9.009$, $df = 6$, $p = 0.173$).

# DISCUSSION

Our results of testing the effect of enrichment in four zoo-housed mammalian species partially support the assumption that enrichment elicits a change in enclosure use and behavioral patterns. First, as expected, enrichment can increase enclosure use in brown bears. However, it had no effect on the use of enclosure space in the other species, i.e., pine martens, domestic ferrets, and golden jackals. Second, object interaction, activity in general, and locomotion increased in pine martens from pre-enrichment to the enrichment phase and activity and locomotion remained high during post-enrichment. Third, visibility in brown bears did not increase with enrichment, but rather with the number of visitors.

Enrichment resulted in a more homogeneous use of enclosure zones in brown bears—especially in the zones where enrichment was introduced, while no effect was apparent in the other three species. The change in enclosure use in brown bears as a result of enrichment is consistent with the findings described for both brown bears and other species (brown bears, *Soriano, Vinyoles & Maté, 2015*; kinkajous *Potos flavus*, *Blount & Taylor, 2000*; fishing cats *Felis viverrina*, *Shepherdson et al., 1993*). Captive carnivores often show low activity and enclosure use due to inadequate feeding conditions (*Bashaw et al., 2003*), which encourages the inclusion of food-based or other types of enrichment. In the present study, a difference in enclosure use was apparent only in the morning (0800 AM till 1100 AM) when the food-based enrichment was freshly placed inside the enclosure. As the acacia logs filled with food were presented only every second day in the morning, a prolonged and more frequent presentation of the enrichment, as well as more frequent changes in location and selection of various hiding places, might enhance the effect and maintain the desired change in a uniform use of different zones over time.

Unsuitable type or location and quantity of enrichment may affect the successful implementation of enrichment (*Van de Weerd & Ison, 2019*), which may have been the reason that domestic ferrets and pine martens did not change their enclosure use. While the pine martens at least manipulated and inspected the food-based enrichment, the domestic ferrets showed little interest in it. This suggests that the type of enrichment was unsuitable for the domestic ferrets. Different objects that can be used for climbing or digging boxes (*e.g.,* boxes filled with rice in which small objects are hidden) could increase their interest. In the current study, the golden jackals responded to olfactory and food-based enrichment during night only and did not emerge from their burrows during any of the phases, providing clear evidence that enrichment did not increase enclosure use, at least during the day. Even though their wild counterparts show higher activity patterns during night, diurnal movements occur on a smaller scale (*Fenton et al., 2021*). However, the type of olfactory and food-based enrichment used may not have been the adequate type of enrichment to change the activity patterns of this carnivorous species to diurnal rhythms. In particular, the choice of scent seems to play an important role as not all scents achieve the desired goals and, in some cases, can even decrease activity (*Clark & King, 2008*). We used semi-concentrated scents in jute bags, which may have caused the scent to dissipate too quickly, causing the animals to lose interest. Finally, unpredictability and the need to search for food may be better suited to elicit a change in activity patterns in golden jackals, as shown for another opportunistic carnivore species, *i.e.,* the red fox (*Vulpes vulpes*, *Kistler et al., 2009*), where electronic feeders delivering food unpredictably in time rather than conventional feeding seem to enhance activity.

Enrichment can increase object interaction, activity in general, and locomotion in pine martens. Enrichment with extrinsic reinforcement (such as food) seem to procure better and more prolonged behavioral changes than ones with intrinsic reinforcements (*i.e.,* behavior itself; *Tarou & Bashaw, 2007*). The food-based enrichments used in the current study did not seem to be effective in the same way for all study species. Only pine martens showed behavioral changes between phases. However, the timing of food-based enrichment was rather predictable (*i.e.,* the objects were always placed inside the enclosures at approximately the same time), which may be a reason why the enrichment was not a success with all four species. Activity rhythms are highly modifiable with food availability (*Boulos & Terman, 1980*; *Ware et al., 2012*). The activity of pine martens in the wild, for instance, changes seasonally and tends to be bimodal in autumn (*Zalewski, 2001*), which could have been a confounding factor for the changes found for activity and locomotion. This would also explain why activity and locomotion remained high during post-enrichment or even increased compared to the enrichment phase. Similarly, captive brown bears are mainly diurnal with a strong crepuscular component (*Ware et al., 2012*) as compared to their wild counterparts, which exhibit nocturnal behavior (*Kaczensky et al., 2006*). Therefore, the activity in this study is consistent with the activity pattern of other captive bears, but diverges greatly from that of animals living in the wild. Still, enrichment can contribute to improved welfare of the animals as it can reduce stress of zoo-housed/captive animals (*Hansen & Berthelsen, 2000*; *McDougall et al., 2006*; *Poessel et al., 2011*) and lack of species-typical inactivity would imply stereotypy (*Renner & Lussier,*

*2002*). Furthermore, habituation can also influence the effectiveness of enrichment, which emphasizes that enrichment should be slightly adapted each time of presentation to induce animals to use the enrichment for longer periods (*Tarou & Bashaw, 2007*).

Contrary to expectations, enrichment did not enhance visibility of animals. But, for brown bears, visibility increased with the number of visitors in front of the enclosure. This suggests that visitors can also act as a form of enrichment for captive animals (*Rault et al., 2020*). Visitors can have positive (*Bloomfield et al., 2015*; *Hashmi & Sullivan, 2020*), neutral (*Sherwen et al., 2014*) or negative (*Davis, Schaffner & Smith, 2005*; *Hashmi & Sullivan, 2020*; *Larsen, Sherwen & Rault, 2014*) impacts on zoo-housed animals, depending on the type of visual or acoustic interactions with the animals. For instance, meerkats (*Suricata suricatta*) showed fewer social interactions during the zoo closure due to COVID-19 pandemic than during times when visitors were allowed back into the zoos (*Williams et al., 2021*). However, this effect probably also depends on the propensity of the animals to interact with humans, which may explain the discrepancy found between species in the current study. Various bear species (*e.g.*, sloth bears *Melursus ursinus*, Andean bears *Tremarctos ornatus*, grizzly bears *Ursus arctos horribilis*, American black bears *Ursus americanus*, Malayan sun bear *Helarctos malayanus*) seem to be more visible in the presence of visitors (*Bernstein-Kurtycz et al., 2021*), while other species seem to show avoidance behavior (*e.g.*, quokkas *Setonix brachyurus*, *Learmonth, Sherwen & Hemsworth, 2018*; orangutans *Pongo pygmaeus*, *Birke, 2002*; jaguars *Panthera onca*, *Sellinger & Ha, 2005*). However, we have to be careful in interpreting the direction of these effects, as visitors may be drawn to enclosures where animals are visible and active.

Although this study shows some findings that may suggest that food-based enrichment can, to some extent, elicit a change in enclosure use and behavior in certain captive carnivore species, these results should be interpreted with caution and may not be generalizable to all other species and facilities due to the small sample size and inconsistent results. Furthermore, the interpretation in this paper is constrained by the limited time period and the changing season (*e.g.*, start of hibernation in the brown bears). Thus, without repeating the ABA design, it is not possible to attribute any changes to the manipulation. Other potential problems are that (1) only one type of enrichment was used per species, (2) the observations were limited to one zoo only, (3) return to baseline data (*i.e.*, data collected at all three phases of pre-enrichment, during enrichment, and post-enrichment) are available only for pine martens and domestic ferrets, and (3) we did not find similar effects across species.

## CONCLUSION

This study has shown that enrichment can elicit an increase in enclosure use for brown bears and an increase in object interaction in pine martens. The results indicate the importance of evaluating long-term effects of enrichment, as not all species reacted to the enrichment and no prolonged changes in enclosure use or behavior were observed. Furthermore, before introducing enrichments, it is important to consider whether the type of enrichment is suitable to achieve the desired goal and does not foster undesired behavior.

Thus, testing species-specific preferences of environmental enrichment prior to integrating and presenting them in the enclosures, as well as testing their effect on enclosure use and behavior, is crucial.

## ACKNOWLEDGEMENTS

We are grateful to the General Manager Bernhard Lankmaier and the animal keepers of the Cumberland Wildlife Park—Daniel Edelbacher, Dietmar Thannesberger, Julian Moser, Hans-Peter Ettinger—for the opportunity of conducting the project and for their help in providing the enrichment for the animals. We thank 86 visitors of the Cumberland Wildlife Park who participated in this study.

### Funding

This project was funded by the FFG Innovation Voucher #889731 to the Cumberland Wildlife Park. The technical equipment for data collection was provided by the Austrian Science Fund (FWF) Doktoratskolleg (DK) Grant Cognition & Communication to Verena Puehringer-Sturmayr (grant number: W1262-B29). Open access funding provided by University of Vienna. The funders had no role in study design, data collection and analysis, decision to publish, or preparation of the manuscript.

### Grant Disclosures

The following grant information was disclosed by the authors:
The FFG Innovation Voucher #889731 to the Cumberland Wildlife Park.
The Austrian Science Fund (FWF) Doktoratskolleg (DK) Grant Cognition & Communication to Verena Puehringer-Sturmayr: W1262-B29.
University of Vienna.

### Competing Interests

Monika Fiby is Zoo consultant, planner, designer and leads her own company.

### Author Contributions

- Verena Puehringer-Sturmayr conceived and designed the experiments, performed the experiments, analyzed the data, prepared figures and/or tables, authored or reviewed drafts of the article, and approved the final draft.
- Monika Fiby conceived and designed the experiments, authored or reviewed drafts of the article, and approved the final draft.
- Stephanie Bachmann performed the experiments, authored or reviewed drafts of the article, and approved the final draft.
- Stefanie Filz performed the experiments, authored or reviewed drafts of the article, and approved the final draft.
- Isabella Grassmann performed the experiments, authored or reviewed drafts of the article, and approved the final draft.

- Theresa Hoi performed the experiments, authored or reviewed drafts of the article, and approved the final draft.
- Claudia Janiczek performed the experiments, authored or reviewed drafts of the article, and approved the final draft.
- Didone Frigerio conceived and designed the experiments, prepared figures and/or tables, authored or reviewed drafts of the article, and approved the final draft.

## Animal Ethics

The following information was supplied relating to ethical approvals (i.e., approving body and any reference numbers):

Austrian Federal Ministry for Science and Research (EU Standard). Animal Experiment Licence Number GZ2021-0.873.421 as indicated below.

## Data Availability

The raw data for each species considered, brown bear, ferret, pine marten, are available in the Supplemental Files.

## Supplemental Information

Supplemental information for this article can be found online at http://dx.doi.org/10.7717/peerj.16091#supplemental-information.

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
