# Peer review of "Effects of food-based enrichment on enclosure use and behavioral patterns in captive mammalian predators: a case study from an Austrian wildlife park"

_PeerJ, doi:10.7717/peerj.16091_

## Round 0.1 · original submission · Major Revisions

I found this to be a very readable paper and I appreciated the inclusion of multiple less-well studied species. All three expert reviewers found merit in your study but all three note the need for improvement. For two of the reviewers, the suggested changes are quite substantial and therefore I have not engaged in line-by-line editing at this stage. I was very fortunate that both of these reviewers shared very detailed and thoughtful reviews with one uploading a very coherent summary of areas needing improvement and the other uploading an annotated version of your manuscript in addition to their detailed review. Please ensure you review both of these documents carefully in preparing your revision.

Like Reviewer 2, I found the information from the questionnaire to be poorly integrated with the main aims of the observational study. It feels like it is tacked on rather than having been conducted to synthesize two possible impacts of enrichment; benefits for the animals and benefits to the public or their perception of the enclosures.

You make the assumption that uniform use of an exhibit space is ideal but there needs to be some justification for this assumption. Different areas may serve different purposes. If an animal naturally spends a significant amount of its time sleeping and it has a preferred sleeping area, disproportionate use of this area would not be a problem. An animal that paces constantly may use most areas equally but this would not be ideal. So, I think more care is needed here. More information is needed about how placement of the enrichment would be expected to impact exhibit space use. If the enrichments were food based, were the foods sometimes depleted before the observations occurred? How did this impact your observations?

Why were the observation periods so unbalanced? How did you account for this in your statistical models? How were the time intervals balanced within the phases? Who did the observations? How were they trained to be reliable and to recognize the individuals?

More detail is needed about the questionnaire. Did you base your questions off of existing questionnaires or did you attempt to re-invent the wheel by creating your own ad-hoc measure? Why did you not match the outcomes of the survey to the enrichment intervals to test whether visitors had better impressions during the enrichment phase, for example? If you did so, you could better integrate these very disconnected aspects of your study. I don’t really see the utility of the visitor survey as presented. I am not sure what it adds or what predictions you had for its inclusion. Nothing is manipulated. As a purely descriptive measure, it is difficult to make use of without a better sense of whether these perceptions are based on particular features of the exhibits.

How is visitor flow defined?

Was subject/species not included in your GLMM?

Why did you use binomial error structure?

What exactly are your hypotheses for the post enrichment phase?
Bernstein-Kurtycz et al. (2021) have also shown that bears are more likely to be visible when more visitors are present, which you cite in the discussion. However, we have to be careful in interpreting the direction of these effects as people may be drawn to exhibits where animals are visible and active. Did you record visitor numbers as how many were present during the observation or just as visitor number to the zoo that day?

As with many other ABA designs, your manipulation is confounded with the passage of time and seasonal changes, which in turn are confounded with visitor activity. Without a repetition of the ABA design, it is virtually impossible to attribute any changes to the manipulation. This limitation at least needs to be addressed. The fact that you didn’t find similar effects across species limits further the generalizability of the claims.

On line 373, you mention three species but just below you mention, bears, pine martens, ferrets and jackals – four species.

Animals often show interest in novel enrichment initially but then quickly lose interest. Would it be better to change the nature of enrichment every day or every few days?

Please adopt a consistent style for references – sometimes you capitalize all words in the titles and sometimes you do not, for example.

Figures 4-7 could be combined.

Reviewer 1 ·

Basic reporting

"Effects of food-based enrichment on enclosure use and behavioral patterns in captive mammalian predators: A case study from an Austrian wildlife park" detailed a straightforward study examining the effect of introducing enrichment into the enclosures of several species to observe whether there was an effect on the behavior and exhibit use. A guest survey on visitor perceptions was also administered.

Overall, the paper is written in a clear manner. The background literature was sufficient and relevant to the research.

Figures and tables were easy to understand. The raw data were included.

Experimental design

The research design was well-defined and relevant to the research question. A few potential improvements/suggestions:

1) Clarify the timeline for the questionnaire. Were these administered throughout the enrichment timeline? (e.g. or were surveys given during enrichment portion only?)

2) Who served as the observers? Report inter-observer reliability.

Validity of the findings

The Chi symbol in the results section is showing as a box.

Report effect size for all significant p-values.

Overall, the authors did a good job not overstating their findings and were clear about the limitations of the study design.

·

Basic reporting

Basic reporting is overall good, but there are several areas that need improvement (some grammar and word choice problems, a suggestion for added literature to the introduction, suggestions for improving and adding figures, and some confusion with the raw data). Importantly, the submission appears to contain two separate experiments (and experimental questions), with the results to these experiments only discussed in conjunction to each other once.

The paper has multiple minor word choice and grammar errors. Commas are occasionally used incorrectly or confusingly (for example, Line 130 “(food-based and sensory, i.e. olfactory, items)”). It is not clear if this is meant to mean olfactory and items or just olfactory items. Other examples for language use and grammar are highlighted in the attached annotated PDF (Lines 68, 124, 125, 130, 167, 185, 186, 193, 197, 241, 260, 319, 337, 362, 365, 388).
Another grammar error throughout is the use of “less.” “Fewer” should be used for countable items (i.e., they always come in wholes, like people and interactions), while less refers to all other items where it would be appropriate to say, for example, there is 1.5 of something (e.g., volumes and amounts). For example, it is not appropriate to say “less than 10 visitors” and should be “fewer than 10 visitors” (Line 332).
The introduction is well organized, detailed, strong, and makes a good case for the authors’ hypotheses. It would be useful to include some references to previous studies of enrichment on these particular species as well as the current literature cited.
The paper conforms to the standard PeerJ structure. Figures are relevant but Figures 1-3 are confusing to interpret. Figures 1-3 show the enclosure use and activity of three of the species studied. While useful, the figures themselves can be hard to look at due partially to the fact that many areas were low use (and thus “white” on the density scale). Unfortunately white is also the color of non-enclosure areas. Borders of enclosures should be made more clear. In addition, background colors are unnecessary and add to the difficulty of interpreting the figure. Readers do not really need to know location of green spaces (with text too small to read) or other obstacles/items in the enclosure in such detail. If the authors still wish to include these items, consider including them all in pale gray-scale. Figures 2 and 3 in particular suffer from background items (in brown, black, gray, and green) making it hard to visualize enclosure use and activity. Another option would be making these background items all low saturation and opacity and adjust the density scale to be higher opacity for even low (but not zero) densities.
I would also appreciate figures or tables that showed the results of the models beyond just the statistical tables (which are useful to include but not sufficient). For example, a table or graph that shows levels (with original numbers) of object interactions, activity levels, and etc. by phase and time interval and highlights which are significantly difference. While these findings can be interpreted from the tables for someone accustomed to the statistical analyses used, it is not immediately obvious.
Raw data is available for the enrichment experiment, but some appears to missing or the labeling scheme is unclear. For example, for the brown bear raw data, observation numbers 2 and 3 are not included, nor are 5, 6, and 7. Numbers of observations labeled do not match the number reported in the manuscript (e.g., 95 observations of brown bears, 79 for pine martens, and 109 for ferrets) while the data sheets label up to 146 for brown bears, 86 for pine martens, and 112 for ferrets. If some observations were excluded, please describe exclusion criteria in the paper. (Note, the authors do say that November data was excluded for brown bears, but the observation numbers prior to November suggest 121 observations, not 95).
In addition, raw data is missing for the visitor questionnaire.
Last, the paper is self-contained, but represents two separate research questions with results that do not overlap and are only once discussed in reference to each other. The first is an experiment on the effects of enrichment on the four captive species. The second is a study of visitor perceptions of these animals and their enclosures. While theoretically related, the authors do little to link together the findings of these two experiments (the only place I could find was Lines 449-450). In addition, the authors somewhat inappropriately link their experiments together by suggesting that visitor opinions may be useful in designing enclosures for animal welfare (Lines 472-473). While visitors may notice non-naturalistic enclosure designs, this does not mean that visitors will also be capable of identifying enclosure designs with high animal welfare. Naturalistic often does create better welfare (but not always, as the authors themselves cite in Line 452), but experts in the species (e.g., those who can identify what is actually natural and not just what looks natural) and in zoo design are still undoubtedly the group that should be consulted for enclosure design, not zoo visitors.
The authors could do more to link together their findings, especially given that they surveyed zoo visitors at the same time period that animals were being observed for the enrichment experiment. For example, did zoo visitor opinions change over the course of the enrichment experiment phases (e.g., pre-enrichment, enrichment, and post-enrichment)?
Last, in-text citations do not follow PeerJ formatting (they do not include the comma between author name and year).

Other minor points of confusion or suggests for rewording throughout are identified in the annotated PDF (Lines 192, 210, 233, 237, 385, 409, 411, 428, 454).

Experimental design

The article represent original primary research within the Aims and Scope of the journal. While the authors describe their research as a case-study (which is not a type of article that PeerJ considers for publication), it is still a research/experimental design (albeit small n). Overall, the research question is well-defined (though a gap in the literature is not) and the method is sufficient but lacks many details necessary to replicate.

The research question is well-defined (what is the effect of enrichment on enclosure use, behavior, and visibility of four captive species at the Cumberland Wildlife Park?), but the gap is not clearly identified. The authors could go into more detail about previous research with these four species and results relevant to their particular aims. For example, the authors discuss previous research in regards to enclosure use in Lines 72-78 but give just four citations, presumably as examples. It is implied that there has not been previous research on how enrichment affects enclosure use, behavior, and visibility in any of their study species, which seems possible for some of their species and measures. However, the authors actually cite a paper on enclosure use in brown bears in the Discussion (Line 385), which should be included in the Introduction. Even still, most zoo enrichment research (like the study here) are small n and thus there is always a need to replicate such designs in different zoos with different individual animals; I find it likely that this study still contributes meaningful information to the field of zoo animal enrichment, but the authors should include a discussion of a gap that they fill and cite additional literature related to their aims and species.
The manuscript lacks any statements about ethics or ethical approval. If these count as “experimental animals” per PeerJ guidelines, then more details on care, feeding, and housing is needed.
The study has a strong design (A-B-A), but the implementation is not particularly rigorous. That said, I consider the main problem to be that discrepancies in the design are not explained (rather than that they exist), as the issues do not limit their analysis, results, or conclusions. For example, in the Study Design section (Lines 161-182), the authors describe the lengths of phases in intervals (e.g., “between 10 and 12 days long” Line 163). Why were the lengths of these phases varying (and presumably the variation was between species, but this is not explained)? Why were the number and length of observations different between phases (within species) and between species? The authors need to justify these differences (either as choices or as due to restrictions in keeper or experimenter availability, etc.).
The methods also lack necessary detail in other areas:
• In Line 165 the authors state that enrichment was placed at low-use/high visibility areas, but no further details are given. Was this the same low-use area each time?
• In Lines 195-196, the authors state that enrichment was renewed every second day without explanation. It is not clear if observations were made on both first and second days, though the authors state in the Discussion that “a difference in enclosure use was only apparent in the morning… when the food-based enrichment was freshly placed inside the enclosure” (Lines 389-390). This could imply that observations were only made on the first day of enrichment, but it is not clear. Nor do the authors justify this timeline (for example, did it take the animals two days to make use of the enrichment? Or some other reason?). In addition, the reason for the jackals’ food enrichment being renewed only once a week is not explained.
• In Lines 196-198, the authors explain why the jackals scent enrichment was only renewed once a week, but do not provide any literature supporting their justification and indeed use wording that implies that the authors had no idea how long scents would be detectable by the jackals (demonstrating a lack of rigor). If the authors did have previous experience, personal communications, or previous literature that led to this choice, they should state it here.
• Line 199 shows a lack of specificity. “The brown bears received two 0.5 m to 1 m long acacia logs.” It is unclear if one log was 0.5 m and the other 1 m or if both logs were somewhere in between (and if so, why not measure them and give exact measurements?).
• Lines 206-209 do not adequately describe the enrichment item for pine martens and ferrets. It is not clear how animals used their paws to retrieve food from a wooden box covered in wire mesh. Was there an opening somewhere in the box? Did the wire mesh cover this opening and was it big enough for paws to reach through, or was the opening in both the mesh and wood? What was the point of the mesh?
• Line 216 the authors state “A pumpkin with food inside seemed close enough as a food-based item” as enrichment for the golden jackals. This is not sufficient justification and suggests that the authors are guessing (demonstrating lack of rigor). Is it true that the authors had no idea whether jackals would like pumpkin, mealworms, or stick insects? This seems like information that could be either probed in a pre-test, found in literature on jackal care, or found in personal communication to other facilities that house golden jackals. I suggest that authors include one of these sources to support the use of pumpkins filled with mealworms and stick insects. At the very least, did the jackals eat any of these items in the course of the study (justifying their use post-hoc)?
• Lines 220-221: How many bags were on the ground and how many 1 m above the ground?
• Line 223 (Measuring enclosure use and behavioral patterns): How were days on which observations collected decided? Were these every day that enrichment was provided, every day, or some random number of days within the phases? In addition, how many people were used as observers? How were they trained? If more than one observer, were any attempts made to code interobserver reliability (i.e., having the observers scan sample the same data to compare for reliability)?
• Line 234: Is interaction with enrichment items part of object interactions? If so, is it differentiated from other object interactions? If not, why not?
• Line 240 (Questionnaire): When and how was the survey administered? As visitors exited the park? At each enclosure? During observations or other times? How did they complete the Excel survey: on their own computers or phones? On a provided laptop or tablet?
• Lines 258-259: The point about jackals using the enrichments should be moved to the Results, but the authors could add the point that jackal results will be discussed descriptively instead.
• Line 262 (Enclosure use): It looks like data was analyzed per scan for behavioral patterns, activity, and visibility, while SPI was entered only once per phase and time interval (making it impossible to do statistical analysis). Given that observations are short (though each includes multiple scans) and thus species are only using a very limited part of the enclosure in each observation, this might be a fair way to do the analysis. However, I wonder if you could calculate SPI per observation or per day (when you have multiple observations on a given day) in order to statistically compare the phases? E.g., on average each day SPI in phase 1 was… Day seems like the relevant unit for enclosure use, rather than the arbitrary week or two that was included for each phase. Or would the small number of observations per day mean that SPI calculated per day was not really representative of enclosure use each day? I’ve never used SPI myself, so I defer to the authors on the best way to use/calculate it. In addition, the difference in enclosure use is visually clear between phases and times of day for the bears, so addressing this is not critical.
• Line 277 (temporal behavior patterns and activity budget): It would be useful to remind readers again which behaviors were measured (in a list, briefly). In addition, please be clear that separate GLMM’s were run for each species and behavior.

Validity of the findings

As discussed in Experimental Design, it is not clear the degree to which this article replicates any previous research, but if it does, it is likely a meaningful replication (presumably including new individual animals in a different location). All data for the enrichment experiment have been provided and results are generally well-described, robust, statistically sound, and controlled. A symbol used in the Results section in multiple places does not appear to have been converted to PDF and appears as □.

There are several places in the Results section where clarity is needed:
• Line 328: The author state that in pine martens, object interaction “increased during enrichment and declined in the afternoon.” Does this mean that objects interactions increased overall during the enrichment phase and (also) was generally higher in morning compared to afternoon?
• Line 329: Did activity increase from pre-enrichment to enrichment and then continue to increase in post-enrichment? Or were enrichment and post-enrichment the same for activity?
• Line 362: Authors report a “good share” of visitors reported “I do not know” to questions. Please report the actual number.

The discussion section is generally well-written and includes good elaboration and speculation on results but has several major points where clarification is needed, where statements in discussion do not match the reported results, and where unwarranted conclusions are drawn.
Conversely, I also think the authors understate what makes their findings interesting. For example, the fact that brown bear enclosure use increased due to enrichment items is worth stating without such strong statements of caution. Instead of saying “the results need to be interpreted with caution” (Lines 375-376) the authors can just state their results more cautiously. For example: “Enrichment can increase enclosure use in brown bears,” is sufficiently cautious. It is important to state limitations, such as small sample size (stated), limited time period (not stated but should be mentioned), and only one type of enrichment used per species (not stated but should be mentioned) but I would not consider the fact that effects were not found in all species to be a limitation unless one had a priori reasons to expect that all species would respond the same to enrichment. Last, the limitations don’t mean that the results must be interpreted with caution (which is vague), they specifically mean that the results may not be generalizable to all other species and facilities. I believe making these changes will make the paper’s discussion and conclusions stronger and more accurately represent what is interesting and worthwhile about the paper (and what its limitations are).
The same is true for the Conclusions section (in Lines 464-466). Here, the authors should highlight their findings of interest (that enclosure use was increased for brown bears and object interaction for pine martens) and note that enrichment can increase these things. Likewise, the authors should state limitations more accurately (small sample size, limited time period, only one type of enrichment used per species, limited to just one zoo/facility, and not the inconsistent results).

The discussion section also has multiple points that need clarification or make unjustified conclusions. Importantly, the authors do not describe in the results or discussion whether the increase in enclosure use for brown bears was in or near the location of the enrichment, which is a key piece of information relevant for anyone who may want to make use of this study (i.e., does enrichment increase enclosure use generally or does it increase enclosure use near the enrichment item?). The authors otherwise do a good job discussing the increased enclosure use by brown bears (Lines 383-393).

On Lines 379-380, and discussed in Lines 432-446, the authors note that brown bear visibility increased with number of visitors. They immediately suggest that the bears made themselves more visible because the visitors were perhaps enriching. However, they do not consider the alternative explanation that more visitors stayed around when the bears were more visible. Unless the authors can report the order of visibility and visitor numbers (e.g., if more visitors appear and then the bears became more visible, this supports their interpretation, but if the bears are more visible and then more visitors show up this supports the alternative), they cannot determine the causal direction of this relationship. The speculation is fine, but the alternative explanation needs to be included as well.

On Line 330, the authors state that pine martens showed higher locomotion post-enrichment compared to the other two phases. Then, in the Discussion on Lines 378-379, the authors state that locomotion increased from pre-enrichment to enrichment. Which of these statements is correct?
Related to this point, the authors state in the Discussion in the same lines that activity increased from pre-enrichment to enrichment, which was true, but fail to note that it either then stayed the same or continued to increase in post-enrichment. This makes it impossible to state that enrichment caused the increase in activity or if some other factor caused the activity increase (e.g., the changing season, as mentioned in Line 422). Likewise, if locomotion indeed only increased in post-enrichment then this change cannot be attributed to enrichment. The point of an A-B-A design is to control for changes over time, and any changes in behavior or measure that do not return to pre-enrichment levels cannot be attributed to enrichment. It may indeed be the case that enrichment simply had long-lasting effects on behaviors, but other factors (again, season being likely) cannot be ruled out. These findings are repeated in Lines 414-415 and again the authors conclude that enrichment had an effect, which is not clear from the stated results. In addition, was object interaction just interaction with the enrichment item, or also with other objects? It’s not clear if the enrichment just inspired interaction with itself, or with interaction generally. Last, with regards to the pine martens, the authors could again state their results more positively. E.g., “enrichment can increase object interaction in pine martens.”

On Lines 381-382 (and Lines 447-449), the authors state that more than half of visitors reported that the brown bear, jackal, and pine marten enclosures looked natural. However, in the results, they report that only 49% of visitors found the pine marten enclosure natural (Line 356), which is much closer to that reported for ferrets (43%) than the other two species (88% for brown bears and 88% for golden jackals, Line 355). Line 382 also states that visitors report that the ferret enclosure would benefit from changes in design, but this was not strictly measured (are the authors averaging data from the multiple questions on enclosure design? The visitors did not think it needed changes in terms of resting spaces, if “needing changes” mean that less than half of visitors said “yes” to the question (with 73% saying the enclosure had enough resting places, Line 358). The Discussion needs to be consistent with the Results.

A minor comment on the statement in Lines 394-395. This statement seems to unnecessarily (and somewhat inaccurately) qualify the findings about the brown bears. It points out that carnivores sometimes adapt better to captivity than other species, possibly due to their small home ranges and short travel distances. I do not think this statement is necessary (first, because “adapt well” does not mean these species have ideal welfare). Nor should the authors leave the implication that encouraging more movement or space use is unnecessary for carnivores because, despite relatively smaller home ranges in the wild than other species, enclosure sizes and distances traveled in captivity are still greatly decreased for almost all captive carnivores

The authors briefly mention behavioral diversity in the Introduction (Line 97), but then focus in on increases or decreases in particular behaviors (rather than behavioral diversity). They do not discuss hypotheses related to or methods that would measure behavioral diversity. While they do record presence and absence of different behaviors, they do not quantify these to get a score of behavioral diversity for any species. It is important to note that this is how behavioral diversity is usually defined (i.e., number of behaviors used in a given observation or scenario). Nevertheless, the authors discuss their results as pertaining to behavioral diversity in Lines 450, 465, and 472. They seem to be using behavioral diversity to refer to changes in rates of behaviors (which is what they measured), rather than number of behaviors displayed, which is an inaccurate use of the term. Unless observed behaviors changed from 0 to a non-zero number, they should not discuss behavioral diversity and instead use the term from their aims and predictions (e.g., just “behavior” or “activity”, Lines 130 and 140-142).

The authors state in Lines 458-459 that “introducing environmental enrichments is valuable to maintain visitor interest” but did not actually measure this in their study (nor are other studies cited for this statement). In the same statement on Lines 460-461, the authors state that these things stimulate visitors to read enclosure signage without citation (and was not measured in this study). Citations from the introduction (or elsewhere) should be included here to make it clear that these are not points that the authors are making based on their own work.

Last, the authors end their paper with the statement that “involving zoo visitors in decision-making processes about animal enclosure design and consequently animal welfare, could raise awareness of nature and promote education” to be unwarranted and I address in my comments for Basic Reporting.

Additional comments

Overall, this is an interesting contribution to the literature on captive animal enrichment and I consider the results worth publishing. It has no fundamental flaws (such as in the method or design) that would warrant rejection, but instead the minor flaws are numerous enough to warrant major revisions, including lack of some critical details in methods, contradictions between results and discussion, conclusions that do not follow from results, and lack of connection between two research goals and methods (the experiment on animals vs. the questionnaire used with humans).

·

Basic reporting

See attachment.

Experimental design

See attachment.

Validity of the findings

See attachment.

Additional comments

See attachment.

---

## Round 0.2 · Minor Revisions

Two of the three original reviewers had a chance to review your revision and found the manuscript much improved. However, both provided further helpful feedback. Notably, both would like you to reimagine how you present the data visually and offer valuable suggestions in that regard. I agree with them that it is preferable to present that information in the main text rather than as supplemental files. I have a few additional requests of my own.

1. Your abstract still refers to the questionnaire (lines 43-44)

2. Line 100 change to say “after climbing structures were introduced.” Otherwise, it reads like the bears introduced the structures to their exhibit.

3. Line 102 is missing a . at the end of the sentence.

4. Line 104 say “ can increase” – positively affect is subjective and it might be unclear what is meant.

5. Insert commas after clauses, e.g., and i.e.

6. Lines 262-263, be very clear if jackals’ data were excluded only in the post hibernation phase or in all phases. Be sure to note in the limitations section that you have return to baseline data only for half of your study species. You should address the finding that the jackals did not emerge from their dens during any of the phases, clear evidence that enrichment did not increase the use of space during the day at least.

7. With time interval, I thought you meant the four-minute intervals throughout the scan but then in the results, it seems like you mean morning, midday and afternoon – please clarify how each variable is operationalized. Perhaps refer to this as time of day instead?

8. Please be more transparent about how you calculated the outcomes – was it just a sum of whether the behavior occurred at all in a given interval regardless of how many animals were engaged in that behavior?

9. Please use the symbol for Chi square rather than writing it out.

10. It might be good to create a figure like 4b in the following paper where you show the stacked behaviors by phase for each species. https://www.researchgate.net/publication/337703105_Time-activity_budget_of_urban-adapted_free-ranging_dogs

·

Basic reporting

The submission uses clear and professional English throughout (though see highlights in the annotated PDF for one word and one sentence that need editing, plus some comma use). Includes sufficient background and relevant literature appropriately referenced. Structure is acceptable.

Figures are relevant but some captions need to be updated from the previous version. New supplementary figures S1-S7 are very helpful (could they be included in main text and not as supplements?). The caption for figure S2 could also include a reminder that, for SPI, 0=equal enclosure use and 1=unequal use. The new figure S8 might not be useful to include: it looks like it shows changes from pre to post-enrichment for visibility, but the only significant effect on visibility was visitor flow (which is not shown in the figure).

The submission is self-contained, an appropriate unit of publication, and includes all relevant results. However, on Lines 37-38 and lines 43-47 the visitor questionnaire is still mentioned (in the abstract).

Experimental design

The paper presents original primary research within the Aims and Scope of PeerJ. The research question is well-defined (effect of enrichment on enclosure use, behavior, and activity of four carnivore species at Cumberland Wildlife Park) with a gap in the literature more clearly identified. The investigation is generally conducted rigorously and following ethical guidelines and methods are now described in sufficient detail to be reproducible. The additional information about number and timings of observation is good.

However, the authors did not complete a measure of interobserver reliability (something that is standard to complete with observational methods). They state in their response to my first review that: “Reliability was trained before starting data collection by defining and discussing the variables (i.e. behaviours) to be collected. This happened at a theoretical and practical level, i.e. in front of the enclosures.” This information should be included in the manuscript as well, with somewhat more detail (e.g., something about how all observers reportedly agreed upon the observed behaviors at this time). The authors do note that they used multiple observers for the brown bear enclosure as one person couldn’t view the entire enclosure. Were there any areas that two or more observers could see at the same time? If so, that data could be used to calculate interobserver reliability at least for the brown bear behaviors. Otherwise, mention of the lack of interobserver reliability measures and what reliability training was conducted is sufficient.

Overall, the authors have greatly improved the paper with the needed extra details for the method.

Validity of the findings

The benefit to the literature is clear: offering new information on what types of enrichment have what kinds of effects on these four species. The paper is not a replication. Underlying data are provided and analysis is well conducted. Conclusions are much improved: highlighting the significant results and discussing actual limitations. It is unfortunate that a goal of the paper was to see if there were long-term effects on behavior post-enrichment but that the findings of increasing activity and locomotion in the pine martens post-enrichment was confounded by seasonality (a factor known to increase their activity and locomotion), but the authors do a good job of discussing this and the need to repeat the ABA design in other time periods.

Additional comments

Overall, the authors have done a great job at improving the manuscript and have now submitted a strong paper that makes an excellent contribution to the overall literature on captive animal enrichment. I recommend accept with minor revisions (a few wording choices, updated figure captions, and some additional information about interobserver reliability).

·

Basic reporting

Overall, the authors adjusted their paper according the the edits I suggested, so I have few concerns with the specific edits. However, I still find following the results based on the way the information is presented difficult, and part of that has to do with a lack of effective visual presentation of those results. It would seem some type of bar graph or activity budget comparing the conditions (A-B-A) should be shown in the main text, not as supplementary material. I think most of the enclosure use information can be left to the text, given that little change in SPI was observed between the conditions. As I mentioned previously, I want to see these behavioral changes. You can move any visual presentation away from enclosure use, because ultimately you found little with respect to changes in enclosure use variability (SPI) based on the conditions. It is also difficult to assert that there were changes in the brown bear's enclosure use, given that this appears to be based on visual inspection with no supporting inferential statistics to support that claim. Nonetheless, my main point here is that some type of visual presentation of the activity of the animals should be presented within the text, and not simply left to supplementary material. I think it would help the reader immensely in following the results if they were able to go beyond a table of results and see graphs with statistically significant results listed in the graphs.

Experimental design

No comment.

Validity of the findings

No comment.

Additional comments

No additional comments.

---

## Round 0.3 · Minor Revisions

Thank you for continuing to undertake the work to provide clarity to your study. I have only a few minor edits to suggest and then I would anticipate being able to accept the next version. All comments with line numbers refer to the tracked version of your manuscript.

Lines 43-44. I don’t like this last sentence of your abstract. You really can’t draw conclusions at the species level here with such small sample sizes. The species differences you observe may actually be individual differences or habitat differences, or seasonal interactions. I would just conclude that effects of food-based enrichment are variable and may have differential effects depending on a variety of factors that would need to be controlled in future studies. You may end on a more positive note by noting that this is one of the first explorations of enrichment use in some of these understudied species. Your last sentence of the first paragraph should indicate the aims of the current study, so you should make a similar statement here after line 74.

On line 78, it seems important to say “but not necessarily equally.” For animals that spend a significant amount of time sleeping, they may not use pools as frequently as shaded areas, or other desirable sleeping areas, for example. I don’t use my laundry room as often as I use my living room or bedroom, but this doesn’t reflect poor design of my home. So, I think line 90 is over-stated.

Line 94, presumably this study involved specific species? It would be better to provide some information here.

The information about reliability is still insufficient. Given the lack of reliability assessments, you should at least be able to indicate how it was determined that observers had met some criteria level of accuracy before being permitted to serve as observers. Reliability means that raters were coding behaviors similarly to how other coders rated those behaviors. Simply defining and discussing the variables does not ensure reliability. If you did training, you should have shown that the trainees indicated the same behaviors that trainers indicated during the same observation. If you do this during training, only then can you describe this as some sort of reliability. Alternatively, you could try to make the case that some behaviors (visible, not visible) and location would be so clear and indisputable that reliability was not deemed necessary. It should be noted as a limitation if only a single coder observed each interval.

The χ2 is not showing up in the PDF. Check journal style but df can be noted in subscript after the χ and test statistics should be italicized.

Line 112 there is an extra “4”

Line 160 change to “Method”

Line 163, License is misspelled.

Line 257 “This references to bears..” is awkward phrasing and not grammatical.

Line 272, place a , after jackal and after “enrichment on line 273 and after cases on line 413.

Line 391, specify “other carnivore species..”

Line 395 “apparent only..” The word “only” is often misplaced. Please check throughout.

Line 415 needs rewording.

---

## Round 0.4 · accepted · Accept

Thank you for making these final small edits to your manuscript.